# Gliomagenesis, Epileptogenesis, and Remodeling of Neural Circuits: Relevance for Novel Treatment Strategies in Low- and High-Grade Gliomas

**DOI:** 10.3390/ijms25168953

**Published:** 2024-08-16

**Authors:** Alessandro Grimi, Beatrice C. Bono, Serena M. Lazzarin, Simona Marcheselli, Federico Pessina, Marco Riva

**Affiliations:** 1Department of Biomedical Sciences, Humanitas University, Via Rita Levi Montalcini 4, Pieve Emanuele, 20072 Milan, Italy; 2IRCCS Humanitas Research Hospital, Rozzano, 20089 Milan, Italy

**Keywords:** glioma, epilepsy, gliomagenesis, epileptogenesis, neurosurgery

## Abstract

Gliomas present a complex challenge in neuro-oncology, often accompanied by the debilitating complication of epilepsy. Understanding the biological interaction and common pathways between gliomagenesis and epileptogenesis is crucial for improving the current understanding of tumorigenesis and also for developing effective management strategies. Shared genetic and molecular mechanisms, such as IDH mutations and dysregulated glutamate signaling, contribute to both tumor progression and seizure development. Targeting these pathways, such as through direct inhibition of mutant IDH enzymes or modulation of glutamate receptors, holds promise for improving patient outcomes. Additionally, advancements in surgical techniques, like supratotal resection guided by connectomics, offer opportunities for maximally safe tumor resection and enhanced seizure control. Advanced imaging modalities further aid in identifying epileptogenic foci and tailoring treatment approaches based on the tumor’s metabolic characteristics. This review aims to explore the complex interplay between gliomagenesis, epileptogenesis, and neural circuit remodeling, offering insights into shared molecular pathways and innovative treatment strategies to improve outcomes for patients with gliomas and associated epilepsy.

## 1. Introduction

Gliomas, the most prevalent primary brain tumors in adults, represent a challenge for neuro-oncology due to their unique characteristics and complex nature [1]. These tumors frequently present with the debilitating complication of epilepsy, with up to 50% of patients experiencing seizures at some point during their disease course [2,3]. The prevalence of epilepsy varies depending on patient- and tumor-dependent factors. Younger age at diagnosis, lower-grade gliomas (oligodendrogliomas and astrocytomas), cortical lesions, peritumoral edema, mass effect, tumor molecular and genetic microenvironment factors, the release of inflammatory mediators, and interactions between the glioma and surrounding neuronal network may all influence local epileptogenicity [4]. Gliomagenesis is the process of development and progression of gliomas, which involves a series of genetic and epigenetic alterations that lead to the transformation of normal glial cells into cancerous cells. These alterations can affect various signaling pathways, DNA repair mechanisms, and cell cycle regulation, ultimately resulting in uncontrolled cell growth and tumor formation [5]. According to some recent research, gliomagenesis and epileptogenesis share common genetic, molecular, and cellular mechanisms (Figure 1). Additionally, the integration of malignant cells and the heightened excitatory transmission in the peritumoral neural network seem to play a crucial role in both glioma progression and seizure development [6]. The intricate relationship between these processes involves overlapping mechanisms and interactions within the tumor microenvironment and surrounding neural networks. Exploring these connections may provide insights into the complex interplay between brain tumors and epilepsy, potentially leading to improved management strategies for patients with glioma-associated epilepsy. By targeting not only the tumor itself but also the underlying mechanisms contributing to epileptogenesis, clinicians may be able to improve patient outcomes, offering a more comprehensive approach to managing this challenging condition.

In this review, we will describe the mechanisms of gliomagenesis and epileptogenesis, explore shared pathways, and highlight novel therapeutic strategies for the management of glioma-associated epilepsy.

## 2. Common Pathways in Gliomagenesis/Epileptogenesis and Novel Target Therapies

### 2.1. IDH 1/2 Mutation

Isocitrate dehydrogenase 1 (IDH1) and isocitrate dehydrogenase 2 (IDH2) are enzymes that play a role in the Krebs cycle and cellular homeostasis. Mutations in IDH1/2 have been found to have a significant impact on various types of human malignancies, including gliomas, acute myeloid leukemia, cholangiocarcinoma, ovarian cancer, and chondrosarcoma [7,8,9,10]. In a study of 1010 diffuse gliomas, IDH1 mutations were detected in 70.9% of the tumors, while IDH2 mutations were observed in 3.1% of the tumors [11]. These two mutations result in neomorphic enzymatic activity, leading to distinct patterns in cancer metabolism, epigenetic alterations, and therapy resistance [12]. The mutant IDH enzymes, particularly the IDH1 R132H and the IDH2 R172 variants, have been extensively studied in lower- and higher-grade gliomas [13,14,15]. These mutations together confer specific enzymatic activity that converts α-ketoglutarate (α-KG) to 2-hydroxyglutarate (2-HG). The accumulation of 2-HG in IDH-mutant tumors profoundly affects various cellular processes [12,16,17].

Elevated levels of 2-HG inhibit α-KG-dependent dioxygenases, including TET enzymes and histone demethylases, resulting in widespread DNA hypermethylation and alterations in histone methylation patterns [18]. These epigenetic changes contribute to the dysregulation of gene expression and promote tumorigenesis. Additionally, the dysregulated production of 2-HG by mutant IDH1/2 alters cellular metabolism, impacting pathways involved in amino acid metabolism, redox balance, and biosynthesis, creating a favorable environment for tumor growth and survival [19]. Furthermore, 2-HG modulates glutamate receptors by mimicking the effect of glutamate at the NMDA receptors and altering synaptic transmission, potentially affecting neuronal network activity and contributing to hyperexcitability [20]. This increased excitability may disrupt the balance of excitatory and inhibitory synaptic transmission, modulate ion channels, and regulate neurotransmitter activity, all of which are implicated in epileptogenesis [21]. Moreover, the close spatial relationship between neurons and glioma cells suggests that the concentrations of 2-HG required to promote mTOR activation in surrounding neurons may be reduced, facilitating direct communication between gliomas and surrounding neurons through direct glutamatergic synapses [22]. The mTOR signaling pathway, which is critically involved in several types of epilepsy, exhibits hyperactivation, leading to increased neuronal excitability despite the absence of histological changes in brain structures [22]. However, the specific mechanism by which mTOR hyperactivation triggers epileptogenesis or seizures is not yet fully understood.

Direct targeting of mutant IDH1/2 enzymes has been a highly pursued strategy in the treatment of IDH-mutant gliomas [23,24,25,26]. D-2-hydroxyglutarate (D-2-HG) inhibitors, such as Ivosidenib (AG-120) and Vorasidenib (AG-881), have shown promising results in preclinical and clinical studies [23,24,25,26]. Vorasidenib is a small molecule inhibitor that effectively crosses the blood–brain barrier and targets both mutant IDH1 and IDH2 enzymes. It is typically administered orally in the form of tablets or capsules [23,24,25]. The INDIGO trial, a double-blind, phase 3 trial, assessed the efficacy and safety of Vorasidenib in patients with residual or recurrent grade 2 IDH-mutant gliomas who had not received previous treatment aside from surgery [25]. The trial demonstrated that Vorasidenib significantly improved progression-free survival and prolonged the time to the next intervention, but the impact on seizures was not specifically reported [25].

Ivosidenib, another IDH1 enzyme inhibitor, has been investigated in patients with advanced IDH-mutated solid tumors [8,23,26]. In a recent multicenter, open-label, phase I trial, Ivosidenib was administered orally on a daily basis in 28-day cycles and demonstrated a favorable safety profile with prolonged disease control and reduced tumor growth in patients with non-enhancing gliomas compared to enhancing gliomas [26]. 

A case study also showed the potential of Ivosidenib in improving seizures in a patient with drug-resistant epilepsy due to an IDH1-mutant grade 2 oligodendroglioma [27]. The administration of Ivosidenib led to a significant decrease in seizure frequency without requiring modifications to the patient’s current anti-seizure medications. This suggests that targeted cancer therapy using Ivosidenib could potentially alleviate the burden of seizures in patients with treatment-resistant epilepsy associated with IDH1 mutant oligodendrogliomas. However, despite promising results, further research is needed to validate the potential role of Ivosidenib and Vorasidenib in attaining improved seizure control [28].

#### PARP-Mediated DNA Repair

PARP (poly-ADP ribose polymerase) is a family of enzymes that play a crucial role in various cellular processes, with a particular focus on DNA repair. One important pathway involving PARP enzymes is the base excision repair (BER) pathway, which is responsible for repairing single-strand DNA breaks [29]. These enzymes detect DNA damage and facilitate the recruitment of other repair proteins to the site of damage, aiding in the repair process. 

PARP inhibitors are substances that block the activity of PARP enzymes, thereby impeding the repair of single-strand DNA breaks. In the context of IDH-mutated gliomas, these tumors heavily rely on the PARP-guided BER pathway to maintain genomic integrity when exposed to genotoxic therapy [30]. By targeting this dependency, PARP inhibitors can induce enhanced apoptotic changes in IDH-mutant cells. Exploiting nicotinamide adenine dinucleotide (NAD+, a coenzyme involved in redox reactions during DNA repair) depletion pathways in IDH-mutant cancers presents a promising strategy to enhance the response to chemotherapy-induced DNA damage and improve the long-term effectiveness of therapy [31]. PARP inhibitors have been extensively studied in various cancer types, particularly BRCA-mutated cancers [32]. In the case of IDH-mutant gliomas, clinical trials have investigated the combination of PARP inhibitors with other agents, such as temozolomide, to enhance treatment efficacy [33,34,35]. Furthermore, recent research has suggested that PARP inhibition may have potential implications in the treatment of epilepsy, given its role in DNA repair mechanisms and neuroinflammation [36].

### 2.2. PI3K Mutation

Phosphoinositide 3-kinases (PI3Ks) are a family of lipid kinases that play a crucial role in cell survival, growth, and proliferation [37]. They are commonly activated in various human cancers and are regulated by growth factors that signal through receptor tyrosine kinases (RTKs) [38]. When activated, PI3Ks phosphorylate the lipid phosphatidylinositol (4,5)-bisphosphate (PIP2) to generate phosphatidylinositol (3,4,5)-trisphosphate (PIP3), which recruits the serine/threonine kinase Akt to the cell membrane [37]. Akt, in turn, phosphorylates and activates mTOR complex 1 (mTORC1) through both direct and indirect mechanisms. mTORC1 is a protein complex that governs cell growth, protein synthesis, and metabolism in response to growth factors and nutrient availability [39]. By phosphorylating downstream effectors like eukaryotic translation initiation factor 4E-binding protein (4E-BP) and p70 ribosomal S6 kinase (S6K), mTORC1 promotes protein synthesis and cell proliferation. The activation of mTORC1 via the PI3K/Akt pathway is essential for coordinating cellular responses to growth signals and nutrient availability [40].

The PI3K pathway is frequently activated in gliomas, particularly higher-grade gliomas, due to gain-of-function mutations in the PIK3CA gene or loss of the tumor suppressor PTEN [41]. Specific variants in the PIK3CA gene have been associated with peritumoral hyperexcitability and seizures in glioma patients [42,43]. For instance, the H1047R variant of PIK3CA has been linked to increased epileptiform activity on EEG, behavioral seizures, and an imbalance between excitatory and inhibitory synaptogenesis compared to other variants such as R88Q [42]. In a retrospective analysis involving 134 glioma patients with PIK3CA variants, the presence of the H1047R mutation was associated with poorer seizure control. These findings suggest that PIK3CA variation may serve as a biomarker for epilepsy severity and response to treatment in glioma patients.

Targeting the PI3K pathway in gliomas has emerged as a promising therapeutic strategy due to its frequent activation in gliomagenesis and epileptogenesis. Small molecule inhibitors that target different isoforms of PI3K have been developed and tested in preclinical and clinical studies for gliomas [44,45]. However, dual inhibitors that target both PI3K and mTOR have demonstrated enhanced efficacy compared to single inhibitors in preclinical studies [46]. The combination of PI3K inhibitors with other targeted therapies or standard treatments like chemotherapy and radiation therapy is currently being investigated to overcome resistance mechanisms and improve the effectiveness of PI3K pathway inhibition [47]. Advances in precision medicine approaches, including genomic profiling, can provide valuable insights into the molecular characteristics of gliomas and help identify patient subgroups that are more likely to benefit from PI3K-targeted therapies. In the case of gliomas, genomic profiling can identify specific genetic alterations in the PI3K pathway, such as gain-of-function mutations in the PIK3CA gene or loss of the tumor suppressor PTEN. These alterations are indicative of PI3K pathway activation and suggest that targeting this pathway with PI3K inhibitors may be an effective therapeutic strategy. By analyzing the genomic profiles of glioma patients, researchers can identify specific molecular subtypes or biomarkers associated with PI3K pathway activation. This information can help guide treatment decisions and identify patients who are more likely to respond positively to PI3K-targeted therapies. For example, patients with specific PIK3CA mutations, such as the H1047R variant, may exhibit a stronger response to PI3K inhibitors compared to patients with other variants. Furthermore, genomic profiling can help identify potential resistance mechanisms to PI3K inhibitors, potentially guiding the development of combination therapies or alternative treatment strategies to overcome resistance and improve the effectiveness of PI3K pathway inhibition.

### 2.3. BRAF Mutation

BRAF (v-RAF murine sarcoma viral oncogene homolog B) is a gene that encodes a protein belonging to the RAF family of serine/threonine protein kinases [48,49]. The BRAF protein plays a pivotal role in the MAPK/ERK signaling pathway, which regulates various cellular processes, including cell growth, proliferation, differentiation, and survival. Mutations in the BRAF gene, particularly the V600E mutation, result in a constitutively active form of the BRAF protein, leading to continuous activation of the MAPK/ERK signaling pathway [50]. Studies have indicated that the BRAF V600E mutation is also associated with the activation of the mTOR (mechanistic target of rapamycin) signaling pathway in glioneuronal tumors [51]. This suggests a potential interaction between aberrant BRAF signaling and the mTOR pathway in the development of these tumors.

Recent research has suggested a potential role for BRAF mutations in the pathogenesis of epilepsy in patients with gliomas [52]. The exact mechanisms by which BRAF mutations contribute to epileptogenesis are not fully understood. However, it is hypothesized that dysregulation of the MAPK/ERK signaling pathway due to BRAF mutations may lead to alterations in neuronal excitability and synaptic plasticity [52]. Another study demonstrated that the BRAF V600E mutation contributes to intrinsic epileptogenicity in pediatric gangliogliomas by inducing the expression of the RE1-silencing transcription factor (REST) [52], which could lead to the acquisition of intrinsic epileptogenic properties in developing neuronal cells. 

Promising targeted therapies for gliomas with the BRAF V600E mutation include BRAF inhibitors such as Vemurafenib and Dabrafenib, as well as combinations of BRAF and MEK inhibitors [53]. Clinical trials have demonstrated objective response rates and prolonged disease control in patients with diffuse gliomas treated with Vemurafenib [54]. Additionally, impressive responses and prolonged disease control have been reported in patients with high-grade gliomas treated with Dabrafenib [55]. Combination therapy with Dabrafenib and Trametinib has shown efficacy in patients with high-grade gliomas, including cases of diffuse metastatic dissemination [56,57].

### 2.4. Glutamate

Glutamate, the primary excitatory neurotransmitter in the central nervous system (CNS), plays a significant role in gliomagenesis through various mechanisms [21]. Glioma cells have been observed to release high levels of glutamate into the extracellular space, resulting in a microenvironment rich in this neurotransmitter [58]. This excessive release of glutamate can lead to excitotoxicity, a process where overstimulation of glutamatergic receptors, including NMDA receptors, can cause neuronal death. The surviving neurons in this toxic environment may exhibit abnormal network functioning, potentially contributing to seizure activity. Additionally, glioma cells express glutamatergic receptors, such as AMPA receptors, which, when activated by glutamate, can induce intracellular calcium influx and stimulate the system Xc-antiporters [59]. This activation enhances the cysteine–glutamate exchange pathway and leads to increased glutamate release from neighboring neurons. Consequently, excitotoxicity promotes neuronal death and supports glioma expansion [59]. The dysregulation of glutamate transporters in glioma cells is a crucial aspect of gliomagenesis. For example, high-grade glioma cells exhibit reduced expression of EAAT-2, a transporter responsible for clearing synaptic glutamate [60]. This reduction in EAAT-2 expression results in increased peritumoral glutamate levels, potentially contributing to a less-excitable tumor microenvironment and the development of higher-grade tumors. Furthermore, glutamate release from glioma cells is closely linked to the uptake of cystine via system Xc-antiporters. This system imports cystine in exchange for the release of glutamate, which serves as a by-product of cellular cystine uptake. Inhibition of system Xc-antiporters has been shown to attenuate glutathione production and significantly reduce tumor growth in vivo, highlighting the importance of glutamate release in glioma progression [58]. 

Glutamate also plays a crucial role in glioma-related epilepsy. The release of glutamate into the extracellular space, combined with its reduced reuptake by astrocytes due to impaired function of excitatory amino acid transporters like EAAT1 and EAAT2, can result in high extracellular glutamate concentrations. Excessive glutamate in the synaptic cleft between cortical pyramidal cells can bind to post-synaptic AMPA and kainate-type ionotropic receptors, leading to membrane depolarization and excitation that promotes seizures [61]. This neuronal activity can trigger the shedding of the synaptic protein Neuroligin-3 (NLGN3), which activates oncogenic signaling cascades and drives synaptic gene expression in glioma cells, potentially contributing to epileptic activity. NLGN3 acts through the PI3K–Akt–mTOR pathway to induce synaptic gene expression, establishing a feed-forward loop that enhances tumor sensitivity to neural activity [62,63]. Furthermore, the upregulation of SLC7A11 (xCT), a subunit of the cystine–glutamate transporter system, in glioma cells has been associated with tumor invasion, outcomes in GBM patients, and the onset of seizures [64,65].

As far as the tumor microenvironment and regional functional circuits are concerned, further findings in the context of brain tumors and epilepsy also emerged from two studies focusing on different aspects of how gliomas influence the neuronal activity, the peritumoral neurons and the extra-cellular matrix to elicit tumor-associated seizures. In particular, in a mouse model with patient-derived xenografts, brain tumors led to seizures by degrading perineuronal nets (PNNs) that insulate fast-spiking interneurons (FSNs). The degradation of PNNs by tumor-released enzymes increases membrane capacitance, reducing the firing frequency of FSNs, which in turn reduces GABAergic inhibition and contributes to tumor-associated epilepsy [66]. A pathological disruption of the potassium and glutamate homeostasis determined by a subset of peritumoral astrocytes with a depolarized resting membrane potential can also determine an increased glutamatergic drive and decreased GABAergic inhibition [67]. This pathological unbalance results from the release of glutamate by gliomas through the Xc-transporter and from the subsequent neuronal loss, and, as such, it represents a significant contributor to the propagation of peritumoral neuronal hyperexcitability and excitotoxic death. Both studies identified critical mechanisms that exacerbate seizure activity, focusing on different aspects of the same issue. A detailed mechanistic insight into how the degradation of PNNs directly impacts FSN function and seizure development is provided in [66], introducing the novel concept that PNNs act similarly to myelin sheaths, reducing membrane capacitance and allowing for high-frequency neuronal firing, which is crucial for maintaining inhibitory control. The understanding is then refined by linking glioma-induced changes in the tumor microenvironment, including glutamate release and neuronal loss, to the overall decrease in inhibitory tone [67]. Together, these studies complement each other, with the first paper [67] offering a more granular mechanism that can be integrated into the broader framework of glioma-induced changes discussed in the second study. Future research could benefit from combining these perspectives, exploring how interventions that protect PNN integrity might synergize with strategies that reduce glutamate release to more effectively manage tumor-associated epilepsy.

In summary, the dysregulation of glutamate release and signaling in gliomas contributes to the development of epilepsy by promoting excitotoxicity, abnormal network functioning, and hyper-excitability of neighboring neurons. 

Understanding the role of glutamate in glioma-associated epilepsy is crucial for the development of targeted therapeutic strategies to manage seizures in patients with gliomas. Therefore, several approaches have been proposed to modulate glutamate levels, receptors, and transporters to potentially mitigate the excitotoxic effects associated with glioma progression and epileptic activity [59,65,68,69,70,71].

Glutamate receptor antagonists: targeting different subtypes of glutamate receptors can help regulate glutamatergic neurotransmission, reduce excitotoxicity, and potentially modulate tumor growth and epileptic activity in gliomas. Non-competitive AMPA receptor antagonists such as Perampanel and Talampanel have shown potential antitumor effects in preclinical studies involving glioma cells [69,70,72,73], as is discussed in the dedicated chapter. 

Excitatory amino acid transporter modulation: enhancing the function of excitatory amino acid transporters (EAATs), such as EAAT2, can help clear excess glutamate from the extracellular space, reducing excitotoxicity [58]. Restoring EAAT2 expression has been shown to decrease glioma proliferation [74]. Targeting EAATs or glutamate transport mechanisms may offer a promising avenue for further therapeutic development.

System Xc-antiporter inhibition: targeting the cystine–glutamate antiporter system Xc- can reduce glutamate release from glioma cells and potentially attenuate tumor growth [59]. Sulfasalazine, a drug historically used to treat inflammatory bowel disease, has been investigated for its inhibitory effects on the system xC- antiporter in gliomas. Preclinical research has shown promising results with Sulfasalazine in glioma models [71]. However, more clinical trials are needed to evaluate the therapeutic potential of inhibiting the system xC-antiporter in patients affected by this condition [75]. Further research is required to assess the safety and efficacy of these strategies in clinical settings.

## 3. Integration in Neural Circuits

The integration of gliomas into neural circuits is a complex and dynamic process involving intricate interactions between glioma cells and neurons. These interactions have significant implications for tumor progression, neural circuit remodeling, cognitive function, seizure development, and patient outcomes [76,77,78]. Glioma cells have been found to establish genuine synaptic connections with neurons through receptors such as AMPA receptors [6]. These synapses enable direct communication between tumor cells and neurons, facilitating the exchange of signals and information. The activation of AMPA receptors on glioma cells is closely linked to the promotion of tumor growth, emphasizing the functional importance of these synaptic connections in glioma progression [63]. 

Additionally, gliomas exhibit electrical coupling with neurons through activity-dependent calcium currents and gap junction-mediated interconnections [6]. This leads to the formation of an electrically coupled network within the tumor mass, enabling coordinated signaling and functional integration of glioma cells into the neural circuitry. The depolarization of glioma cell membranes, triggered by neuronal activity, has been shown to drive tumor proliferation, underscoring the role of electrical signaling in glioma progression [63]. Gliomas can also influence neuronal excitability and activity in the surrounding brain tissue, potentially leading to activity-regulated glioma growth [6,20,79,80]. The bidirectional interaction between glioma cells and neurons creates a feedback loop wherein tumor-induced changes in neuronal activity further promote tumor progression [6]. 

Furthermore, the presence of gliomas within neural circuits can result in the remodeling of functional neural circuitry, extending the activation of the tumor-infiltrated cortex during specific tasks and impacting neural networks and cognitive functions [79]. As previously mentioned, the synaptic protein NLGN3 plays a crucial role in the interaction between glioma cells and neurons, promoting the proliferation of high-grade glioma cells. NLGN3 released by neurons in response to activity stimulates glioma cell proliferation and induces the expression of synapse-related genes in glioma cells [6,81]. Another protein, IGSF-3, has been identified as a regulator of glioma progression and brain network hyperactivity [82]. IGSF-3’s association with synaptic gene sets enriched in glioma patients experiencing seizures suggests its involvement in brain network hyperactivity. Functional studies have demonstrated that IGSF-3 drives glioma progression through synaptic remodeling and network hyperactivity by interacting with the potassium channel Kir4.1. Disruption of Kir4.1 function can lead to elevated extracellular potassium levels, increasing neuronal excitability and contributing to epileptiform activity [83]. 

Targeting NLGN3 and IGSF-3 could present potential therapeutic strategies to inhibit glioma progression and epileptiform activity. Developing specific antibodies or inhibitors against these proteins, as well as targeting ADAM10, an enzyme involved in the cleavage of NLGN3, may offer promising approaches to disrupt their function and improve patient outcomes. Further research is necessary to evaluate the efficacy and safety of these therapeutic interventions in preclinical and clinical settings.

## 4. Clinical Relevance

Optimal seizure control is a primary objective in the clinical management of patients with low- and high-grade gliomas. In addition to conventional measures such as progression-free survival (PFS) and overall survival (OS), quantifying seizure control provides a more comprehensive assessment of clinical outcomes in glioma patients [84]. It helps evaluate the impact of treatment on symptom management and quality of life, especially when investigating novel compounds or therapies targeting common pathways of epileptogenesis and tumor cell proliferation [84]. The development of tailored medical and surgical strategies is crucial for patients with glioma-related epilepsy, both for oncological and epileptological purposes.

### 4.1. Surgical Treatment and Advanced Imaging Techniques Addressing Gliomas and BTRE

Advancements in surgical techniques have greatly influenced the management of gliomas and BTRE. Studies demonstrated that achieving a maximally safe resection of gliomas is associated with improved disease control, reduced risk of recurrence, enhanced overall survival rates, and better seizure control [85,86]. In this setting, connectomics, a field of neuroscience focused on mapping and analyzing neural pathways in the brain, has played a significant role in enhancing the concept and implementation of safe resection. Connectomics allows neurosurgeons to identify subcortical structures of the brain involved in important neurological functions, such as motor–sensory, visual, and cognitive functions. By detecting these circuits in real time during surgery, it is possible to constrain the risk of postoperative deficits without sacrificing the extent of the resection [87]. 

Connectomics also aids in identifying epileptogenic networks, which are regions involved in seizure generation and propagation [88]. Various techniques, such as diffusion tensor imaging, functional MRI, and electroencephalography, enable clinicians to analyze the brain as a complex functional network, leading to more precise identification and removal of the effective epileptogenic tissue, beyond what is depicted as a lesional area by conventional radiological tools, while preserving critical functional networks [88]. The integration of connectomics into neurosurgery practice has thus the potential to contribute to the management of both intra-axial brain tumors and BTRE. By leveraging insights provided by connectomics, neurosurgeons can perform tailored surgical procedures, resulting in improved outcomes for both oncological and epileptological aspects of patient care (Figure 2). The resulting surgical planning is thus multi-modal, integrating structural and functional data that stem from different diagnostic techniques, and seeks a clinical impact that has oncological, neurological, and functional relevance. 

Concerning surgical techniques, “supratotal resection” has progressively emerged during the last decades in the neuro-oncological field as a safe and effective method to improve PFS and OS in both low- and high-grade gliomas [86,89,90,91]. This technique involves extending the resection of an intrinsic brain tumor beyond its radiological margins. Clinical studies have investigated the safety and feasibility of supratotal resection guided by intraoperative mapping in lower-grade gliomas, and research has shown that supratotal resection leads to improved long-term seizure control compared to partial or gross total resection [86,89,92]. For high-grade gliomas, supratotal resection has also shown potential for improving progression-free survival and overall survival in certain patient populations [93,94]. The concept of supratotal resection highlights the importance of a multidisciplinary approach to brain tumor surgery, involving various specialists such as neurosurgeons, neuro-oncologists, neurologists, neuropsychologists, and intraoperative neurophysiologists. 

Advanced imaging techniques, including functional MRI, diffusion tensor imaging, PET, and SPECT, play a crucial role in the presurgical planning of epileptogenic gliomas [95,96,97,98]. These techniques help identify epileptogenic foci, which can vary depending on the tumor type and molecular profile. The “peritumoral zone”, an area infiltrated by isolated tumor cells, is often the origin of epileptogenic foci in lower-grade gliomas [99]. Various imaging techniques can be used to detect these foci, enabling a better understanding of the epileptogenic network associated with the tumor [99]. For example, diffusion tensor imaging (DTI) allows for the non-invasive study of white matter fiber organization and can indicate alterations in white matter integrity linked to epileptogenic foci [98]. PET imaging, particularly with aminoacidic PET tracers like [11C] methionine (MET), provides insights into tumor metabolism, grade, neuroinflammation, and associated epileptogenicity. MET–PET can help localize epileptogenic foci within or around gliomas by detecting areas of increased amino acid uptake and metabolism [100,101]. By incorporating these advanced imaging findings into clinical practice, neurosurgeons can optimize treatment strategies by tailoring the extent of resection and targeting specific epileptogenic foci. This personalized approach based on the metabolic characteristics of the tumor can lead to improved outcomes in terms of both oncological and epileptological aspects of patient care.

### 4.2. Antitumoral Effects of Antiseizure Drugs (ASDs) and Clinical Management of Seizures

Experimental evidence suggests that certain antiseizure medications (ASMs) may have antitumorigenic effects, indicating that correcting abnormal neurotransmitter activity in gliomas could address both seizure control and glioma cell proliferation and progression [102]. In this setting, promising effects have been observed with valproic acid (VPA), levetiracetam (LEV), gabapentin (GAB), perampanel (PER), brivaracetam (BRI), and lacosamide (LAC) (Figure 3).

#### 4.2.1. Valproic Acid (VPA)

VPA is an antiseizure drug that enhances the level of endogenous gamma-aminobutyric acid (GABA) and blocks voltage-gated sodium (Na^+^) and calcium (Ca^2+^) channels. In addition to its anti-epileptic effect, many studies have documented a survival benefit in glioma patients. The anti-tumor effects of VPA are not fully understood, but possible mechanisms have been reported [103,104]. Preclinical data suggest that VPA alone or in combination with temozolomide (TMZ) may inhibit cancer cell growth by inducing apoptosis, cell cycle arrest, and autophagy [104,105]. VPA also stimulates histone hyperacetylation, leading to an unfolding of the chromatin structure and enhancing DNA susceptibility to the effects of radiation therapy and chemotherapeutic drugs [106]. Furthermore, VPA plays a role in inhibiting histone deacetylases and can suppress the epithelial–mesenchymal transition process, compromising the DNA integrity of tumor cells [107]. The available clinical data support these in vitro findings, showing improved outcomes in glioblastoma (GBM) patients treated with VPA in combination with TMZ [108]. 

#### 4.2.2. Levetiracetam (LEV)

LEV is an ASM that binds to a synaptic vesicle glycoprotein (SV2A), inhibiting presynaptic calcium channels and acting as a negative allosteric modulator of GABA- and glycine-gated currents [109]. Studies have suggested that LEV has clinical benefits beyond seizure control [110,111,112]. Its antitumor efficacy has been attributed to its inhibitory action on the DNA repair protein MGMT, which plays a crucial role in tumor cell resistance [113]. LEV decreases MGMT protein and mRNA expression levels, resulting in the inhibition of malignant glioma cell proliferation and increased sensitivity to TMZ [113]. Retrospective studies have, indeed, shown promising survival benefits in GBM patients receiving LEV- and TMZ-based chemotherapy, especially in patients with methylated MGMT promoter [114,115,116]. However, there are conflicting findings regarding the effectiveness of LEV concerning MGMT methylation [112]. Nonetheless, it is essential to investigate whether MGMT methylation in gliomas can be used as a predictive marker for the effectiveness of LEV, both as an antiepileptic drug and as a treatment for cancer. A recent study by Pallud and colleagues revealed that combining LEV treatment with standard radiotherapy and chemotherapy may enhance overall survival in GBM patients with IDH1 wild-type tumors [117]. This suggests that the effectiveness of LEV as an antineoplastic agent may be influenced by the IDH1 status of the tumor. 

#### 4.2.3. Gabapentin (GAB)

Gabapentin interacts with voltage-activated calcium channels and binds to the α2δ-1 subunits, leading to the inhibition of neurotransmitter release and attenuation of neuronal excitability [118]. Gabapentin has shown promise in inhibiting glioma proliferation through its interaction with Thrombospondin-1 (TSP-1), a protein produced by high-grade glioma cells that promotes tumor invasion [119,120]. In vitro studies have demonstrated that gabapentin reduces glioma proliferation when co-cultured with neurons and decreases glioma proliferation in mice with patient-derived xenografts [79]. In this context, the ability of gabapentin to inhibit TSP-1 and modulate neural interactions may provide a novel approach to targeting glioma proliferation. 

#### 4.2.4. Perampanel (PER)

Perampanel is a non-competitive AMPA receptor inhibitor primarily used to treat BTRE [121]. As mentioned before, this drug seems to have shown potential antitumor effects, particularly in preclinical studies involving glioma cells. These studies have demonstrated that Perampanel can reduce cell metabolism and induce apoptosis in glioma cells, leading to antiproliferative effects [72]. Additionally, Perampanel has been found to have a synergistic effect with Temozolomide (TMZ), the standard chemotherapy for glioblastoma. The exact mechanisms by which Perampanel exerts its antitumor effects are not fully understood, but studies suggest that it may impact cell metabolism, induce apoptosis, and modulate the expression of glutamate receptor subunits to inhibit cell migration and promote apoptotic cell death [122]. However, clinical studies investigating the use of PER treatment in BTRE are limited and have only focused on its efficacy as an add-on therapy [73,123,124]. 

#### 4.2.5. Brivaracetam (BRV) and Lacosamide (LAC)

BRV, like LEV, binds to synaptic vesicle protein 2A in the brain, while LAC enhances the slow inactivation of voltage-gated sodium channels. A study by Rizzo and colleagues showed that these two new-generation ASMs may have an antineoplastic effect on glioma cell lines [125]. In vitro, BRV and LAC demonstrated an anti-migratory effect and dose-dependent cytotoxicity that was not related to apoptosis [125]. Exposure of glioma cells to BRV and LAC resulted in the modulation of several microRNAs, with miR-107 implicated in inhibiting cell migration and miR-195-5p affecting the cell cycle [125]. Furthermore, the BRV and LAC treatment did not affect the expression of chemoresistance-related molecules MRPs1-3-5, GSTπ, and P-gp [125].

In focal structural lesions such as brain tumors, the occurrence of even a single seizure leads to the diagnosis of BTRE. According to the recommendations of the International League Against Epilepsy (ILAE), ASMs can be initiated after the first seizure, and there is no evidence that prophylactic treatment is effective [126,127,128]. Evidence supports the use of levetiracetam, zonisamide, carbamazepine, phenytoin (Class A), and valproic acid (Class B), or gabapentin, lamotrigine, oxcarbazepine, phenobarbital, topiramate, and vigabatrin (Class C) in focal epilepsy. The final choice of ASM is based on individual patient characteristics, such as age, sex, weight, risk of adverse effects, comorbidities, or co-therapies that may increase the risk of drug interactions [129]. The use of enzyme-inducing antiseizure drugs should be avoided in BTRE, as they may interfere with anti-tumor treatment and other AEDs and dexamethasone [126,130]. Considering the literature, the choice of ASM should also take into account its potential antineoplastic activity to improve patient survival and control both BTRE and tumor growth and recurrence. Based on their designation as class A and B agents for focal epilepsy and their antitumor effects, LEV and VPA may be considered first-line treatments for BTRE [131]. In low- or high-grade glioma patients, levetiracetam should be the initial therapy due to its minimal drug interactions, high tolerability, and ability to improve cognitive function in a significant percentage of patients (20–25%) [110,111]. The most common adverse effect of LEV is irritability, which leads to discontinuation in less than 5% of patients [132]. LEV can achieve seizure freedom in 60–100% of cases, although prior surgery and associated antitumor therapy likely contribute to these results [116,132]. Valproic acid can be considered a valid alternative in monotherapy due to its safety and effectiveness [131]. However, VPA treatment is associated with common and serious adverse effects, including nausea/vomiting, somnolence, liver failure, low blood platelets, and congenital abnormalities [133,134]. If monotherapy with an ASM fails to control BTRE, introducing a second ASM as an add-on therapy is recommended. A retrospective study showed that combining levetiracetam with valproic acid achieved seizure freedom in 60% of glioblastoma patients who did not respond to monotherapy [135]. Furthermore, levetiracetam exhibits synergistic effects when combined with GABAergic or glutaminergic activity modulators, enhancing its effectiveness in combination therapy [136]. In the event of therapeutic failure with first-line therapy, alternative options for the treatment of BTRE may include Lacosamide (LAC) [137], Perampanel (PER), or Brivaracetam (BRV) [73]. These drugs are considered due to their effectiveness, tolerability, and mechanism of action, which have the potential to impact tumor growth. In addition, the withdrawal of antiseizure medications (ASMs) is not recommended for patients at a high risk of seizure recurrence, regardless of the duration of seizure freedom [138]. This includes patients with tumor progression, a history of status epilepticus, and highly malignant tumors with a poor prognosis [138,139].

## 5. Conclusions

The intricate interplay between gliomagenesis, epileptogenesis, and neural circuit remodeling highlights the complex nature of both low- and high-grade gliomas. A comprehensive understanding of the shared molecular pathways, metabolic alterations, and structural changes that contribute to tumor progression and seizure activity is essential for the development of novel treatment strategies. 

By targeting common mechanisms involved in glioma development, epileptic seizures, and neural circuit remodeling, researchers can explore innovative therapeutic approaches that address the complex interrelationships between these processes. Integrating insights from the fields of glioma biology, epileptology, and neuroplasticity may lead to the discovery of precision medicine interventions that not only target tumor growth but also regulate epileptic manifestations and restore neural circuitry function. Ultimately, these advancements have the potential to greatly improve outcomes for patients with both low- and high-grade gliomas.

## Figures and Tables

**Figure 1 ijms-25-08953-f001:**
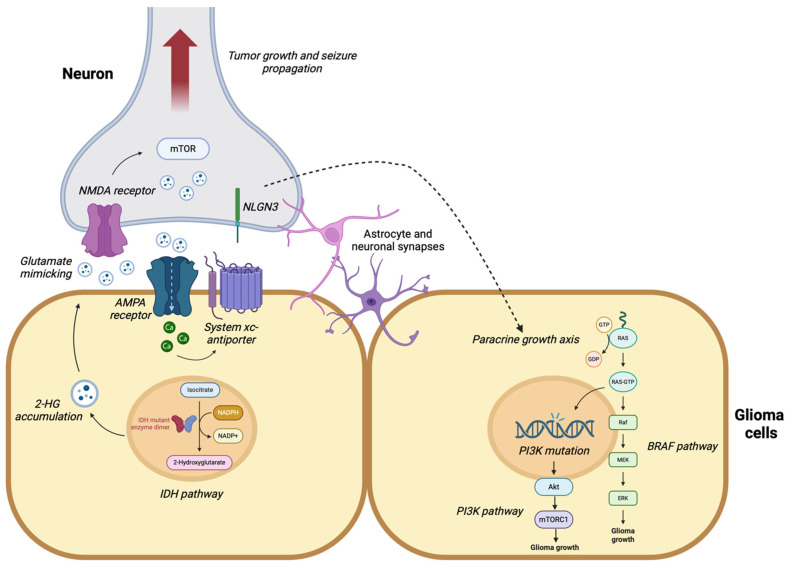
Shared pathways between gliomagenesis and epileptogenesis.

**Figure 2 ijms-25-08953-f002:**
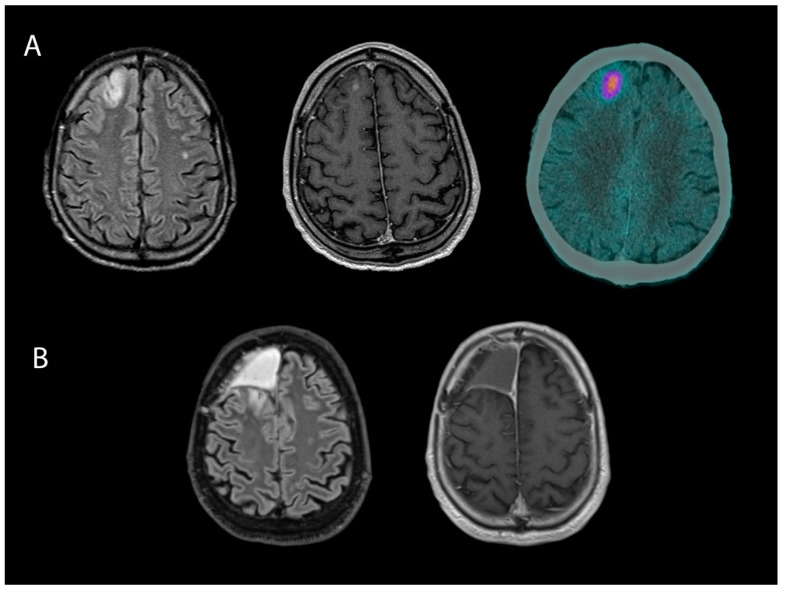
Presurgical imaging (FLAIR, T1-w+c MRI, and MET–PET) of a right frontal IDH wild-type glioblastoma (WHO grade 4) surgically treated at our institution. The patient had uncontrolled seizures before surgery and underwent a supratotal resection guided by neuromonitoring (SSEP, MEP, DES, and ECoG) (**A**). After two years, the patient is still seizure-free, and there has not been a recurrence of the disease (**B**).

**Figure 3 ijms-25-08953-f003:**
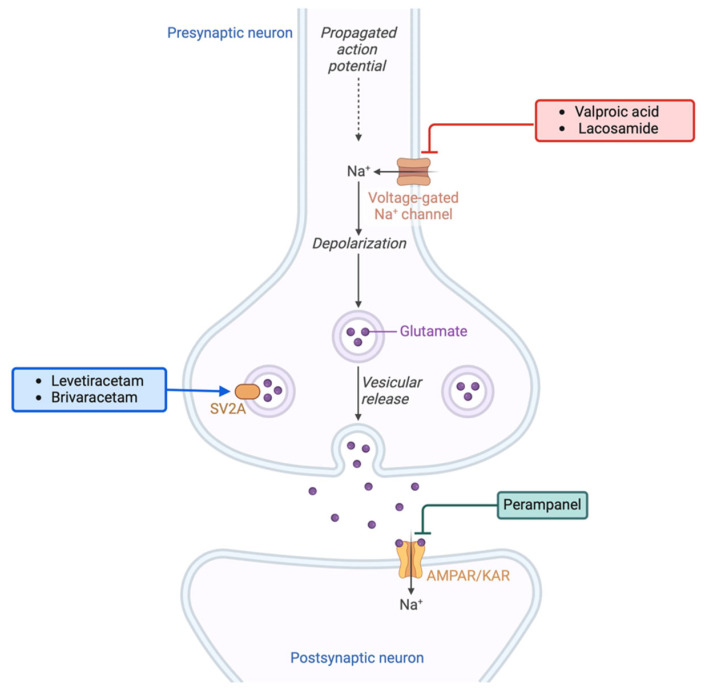
Mechanisms of action of AEDs with anti-tumorigenic effects.

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
