# Peer review of "Gliomagenesis, Epileptogenesis, and Remodeling of Neural Circuits: Relevance for Novel Treatment Strategies in Low- and High-Grade Gliomas"

_ijms, 2024, doi:10.3390/ijms25168953_

Round 1

Reviewer 1 Report

Comments and Suggestions for Authors

There are a number of reports in the literature on the relationship between epilepsy and glioblastoma. In this work, the focus has been on the molecular pathways, metabolic alterations, and structural changes encountered in the progression of the tumor through its different stages and in epileptic activity. Understanding these mechanisms can be helpful for developing new therapeutic approaches, which can improve the quality of life for patients even in a high-grade diagnosis. Figures 1 and 2 are very clear in explaining the molecular mechanisms involved in these processes. I'm sorry, could you clarify your request? Are you asking for additional neuroimages to be added to figure three to enhance its clinical and methodological aspects?

Author Response

We thank the reviewer for the comments and appreciation of our manuscript

Reviewer 2 Report

Comments and Suggestions for Authors

The goal of this review was to clarify the relationship between gliomagenesis and epileptogenesis in order to highlight new therapeutic opportunities for the management of gliomas.

The topic was interesting, although there are already similar published reviews. However, the quality of the manuscript is largely disappointing. The language is correct but the ideas are developed in a very confusing manner. Main ideas are not highlighted. The structure of the manuscript is absolutely not logical. Most references are inappropriate as many reviews are cited while many important studies in the field of cancer neuroscience are not cited and discussed. Aboveall, the authors failed to correctly explain how some molecules are able to target both epilepsy and glioma growth.

1.       Abstract : the scope and goal of the review is not clearly precised.

2.       Line 53: title “Pathways”. Which pathways are considered? Pathways involved in gliomagenesis?

3.       In the paragraph regarding IDH mutations, the authors only consider IDH1 mutations. Could they add data regarding IDH2 mutations?

4.       Line 65 to 81 and line 545 to 546: original references are lacking.

5.       The plan of the article is not very logical and lacks consistency. For instance, paragraph 2.13 is redundant with lines 69-72 and do not bring a lot of additional relevant elements. Additionally, glutamate (part 2.4) is one of the preferential neurotransmitters of neuro-glial synapses (corresponding to part 3). Neuroligin 3 driven mechanisms and perampanel application, for instance are discussed redundantly in these 2 parts.

6.       Line 232: the sentence has to be rephrase. V600E BRAF mutation only concerns a small proportion of gliomas and does not play a so critical role in gliomagenesis.

7.       The authors cite too many reviews and too few original papers (ex: ref 20, 52, 53, 54, 66, 67, 75, 79, 99, 116…).

8.       Line 356: the coupling between neurons and glial cells relies mainly on calcium currents.

9.       Line 411-414: how could CRISPR-CAS9 techniques applied in living humans???

10.    Line 440-441: regarding anti-epileptic drugs that can be repurposed to offer an antimural effect, the author forget some examples as GABAPENTIN (Krishna et al, Nature, 2023).

11.    Regarding Perampanel, this important study is not cited: “Rogawski, M.A., and Hanada, T. (2013). Preclinical pharmacology of perampanel, a selective non-competitive AMPA receptor antagonist. Acta Neurol Scand Suppl, 19-24. 2109 10.1111/ane.12100. »

12.    Paragraph 4.1.4: whereas the paragraph is entitled BRV and LAC, the authors discuss about levetiracetam and valproic acid.

13.     Line 556: the authors cite GABAergic signaling as a target (which is relevant) without any reference (while the are several in the literature) or argument for that.

14.    Paragraph 4.2: as the review is based on pharmacological targeting of gliomagenesis and epilepsy, it is weird to see a surgical paragraph. Additionally, this paragraph comes very late in the review while surgery is the first step of the treatment of glioma. Regarding surgery, why are the surgical results, in terms of epilepsy control, not detailed? The concept of supramaximal resection, whether probably relevant in low-grade gliomas, is strongly put into question in high-grade gliomas. Recent references on the topic are lacking.

Comments on the Quality of English Language

Only minor edits are needed.

Author Response

We thank the reviewer for the comments provided.

We addressed them and we herein report as follows:

  1. Scope of review has been better specified
  2. Rephrased as "Common pathways in gliomagenesis and epileptogenesis"
  3. Data regarding IDH-2 added (from line 59 onwards)
  4. references added
  5. paragraphs modified to increase consistency
  6. sentence rephrased
  7. references edited
  8. argument considered 
  9. sentence edited
  10. GABAPENTIN added as suggested
  11. Reference added
  12. title edited
  13. references added
  14. paragraph edit to be more accurate. 

Round 2

Reviewer 2 Report

Comments and Suggestions for Authors

I regret that the authors only provided very short responses to comments, which is not very respectful to the reviewers’ job.

To me, extensive editing was needed and only minor corrections were made. Consequently, the manuscript is still very confusing and do not reach the level of clarity that could be expected by IJMS readers. For instance, in the second part, I still not see to what extent the authors give a parallel overview to pathways commonly affected in gliomagenesis and epileptogenesis. Another example can be found lines 624-625: supratotal resection including removal of the anterior temporal lobe (AND NOT ONLY ANTERIOR TEMPORAL LOBECTOMY) offers better results in terms of seizure control compared to gross total resection.

Above all, there are already many reviews on the same subject and this one do not give new perspectives, which is yet the essential goal of a well-done review.

Author Response

We employed the Reviewer's Comment of the first round as guide.

Notes to reviewer's comments can be found in the attached document

Round 3

Reviewer 2 Report

Comments and Suggestions for Authors

The quality and the clarity of the manuscript have been widely improved. This review is now very nice and relatively different from previous reviews. The link is clearly established, from fundamental data to clinical practice and patient care.

I have just a few minor remarks:

1.      In the abstract, the key concept that there are complex bidirectional direct interactions between neurons and tumor cells (and not only shared pathways) is lacking.

2.      Regarding glutamate, these 2 important references should be added and discussed.

Campbell SC, Muñoz-Ballester C, Chaunsali L, et al (2020) Potassium and glutamate transport is impaired in scar-forming tumor-associated astrocytes. Neurochem Int 133:104628. https://doi.org/10.1016/j.neuint.2019.104628

Tewari BP, Chaunsali L, Campbell SL, et al (2018) Perineuronal nets decrease membrane capacitance of peritumoral fast spiking interneurons in a model of epilepsy. Nat Commun 9:4724. https://doi.org/10.1038/s41467-018-07113-0

3.      Line 316: “NLGN3's release” must be replaced by “NLGN3 release”. Same for IGSF-3's

4.      Line 342-360 : this paragraph is confusing. It is true that connectomics identifies white mqtter tracts that must be spared during tumor resection and can assess epileptogenic networks. Yet, there is absolutely no links between these two different problems and I do not think it is optimal to mix all these ideas in the same paragraph.

Comments on the Quality of English Language

Fine

Author Response

The quality and the clarity of the manuscript have been widely improved. This review is now very nice and relatively different from previous reviews. The link is clearly established, from fundamental data to clinical practice and patient care.

#We thank the reviewer for the appreciation of our revision. We modified the manuscript accordingly highlighted with Track changes.

I have just a few minor remarks:

  1. In the abstract, the key concept that there are complex bidirectional direct interactions between neurons and tumor cells (and not only shared pathways) is lacking.

#The abstract has been edited to include this comment

  1. Regarding glutamate, these 2 important references should be added and discussed.

Campbell SC, Muñoz-Ballester C, Chaunsali L, et al (2020) Potassium and glutamate transport is impaired in scar-forming tumor-associated astrocytes. Neurochem Int 133:104628. https://doi.org/10.1016/j.neuint.2019.104628

Tewari BP, Chaunsali L, Campbell SL, et al (2018) Perineuronal nets decrease membrane capacitance of peritumoral fast spiking interneurons in a model of epilepsy. Nat Commun 9:4724. https://doi.org/10.1038/s41467-018-07113-0

            #references added and discussed as suggested

  1. Line 316: “NLGN3's release” must be replaced by “NLGN3 release”. Same for IGSF-3's

#edited as suggested

  1. Line 342-360 : this paragraph is confusing. It is true that connectomics identifies white mqtter tracts that must be spared during tumor resection and can assess epileptogenic networks. Yet, there is absolutely no links between these two different problems and I do not think it is optimal to mix all these ideas in the same paragraph.

#The paragraph was edited to clarify the meaning of what was originally reported. We hope to have achieved a high enough level of clarity about an issue we consider relevant in both the current practice of neurosurgery and for their future advancements.